

# Boulder transport and wave height of a seventeenth century South China Sea tsunami on Penghu Islands, Taiwan

Neng-Ti Yu [1]*, Cheng-Hao Lu [2], I-Chin Yen[3], Jia-Hong Chen[4], Jiun-Yee Yen[5], Shyh-Jeng Chyi [6]

[1] Center for General Education, National Tsing Hua University, 101, Sec. 2, Kuang-Fu Rd., Hsinchu City 300044, Taiwan; ntyu@mx.nthu.edu.tw
[2] Department of Tourism and Leisure, National Penghu University of Science and Technology, 300, Liuhe Rd., Magong City, Penghu County 880011, Taiwan; luch@gms.npu.edu.tw
[3] Graduate Institute of Applied Geology, National Central University, 300, Zhongda Rd., Zhongli Dist., Taoyuan City 320317, Taiwan; ichin.yen@gmail.com
[4] Department of Natural Resources and Environmental Studies, National Donghua University, 1, Sec. 2, Da Hsueh Rd., Shoufeng, Hualien 974301, Taiwan; jiahong1017@gmail.com
[5] Department of Natural Resources and Environmental Studies, National Donghua University, 1, Sec. 2, Da Hsueh Rd., Shoufeng, Hualien 974301, Taiwan; jyyen@gms.ndhu.edu.tw
[6] Department of Geography, National Kaohsiung Normal University, 116, Heping 1st Rd., Lingya District, Kaohsiung City 802561, Taiwan, chyisj@nknu.edu.tw

*Correspondence to*: Neng-Ti Yu (ntyu@mx.nthu.edu.tw)

**Abstract.** The widespread tsunami risks in the South China Sea have diverse origins from trench megathrust to intraplate earthquake or landslide and remain poorly understood due to the scarce historical and geological records. The cliff-top paleotsunami gravels and basalt boulders on Penghu Islands in the Taiwan Strait present facies constraints on sediment transport, wave estimates from incipient motion formulas, and stratigraphic links to the probable sources. The boulders are supported by a pumice-bearing mud matrix that reflects a suspension-rich turbulent flow process and the typical rolling–saltation transport that results from bore-like waves. Calibrating for ancient sea level height and 100 year surge indicates that the storm waves that are likely to form in the shallow interisland bathymetry only enable boulder sliding–rolling and are incapable of the 2.5 m high cliff-top deposition. The estimated minimum height of tsunami waves is also insufficient and needs to add to 3.0 m high for a minimum cliff-top overflow of 0.5 m depth for terminal rolling before deposition. Coeval gravels in two other outcrops also record the time and extent of tsunami deposition, and are characterized by beach-derived bioclasts and stranded pumices, sharp base, matrix support, poor sorting, and elevation reaching above the 100 year surge. The gravels mark the local minimum wave run-ups and reach 2.4–4.0 m above sea level. The 1575–1706 radiocarbon age of the studied boulder suggests a probable tie to the disastrous 1661 earthquake in the SW Taiwan Orogen and the megathrust source in the northern Manila Trench.





## 1 Introduction

Devastating wave impacts, such as those associated with the annual typhoon, are a constant threat at the coasts of the South China Sea. The potentially hazardous tsunami waves are, however, often overlooked because of extended recurrence, although ubiquitous and diverse sources such as the Manila Trench in the east, intraplate seismicity in the north–west, and steep slopes with pronounced landslide potential, are found all over the region (Fig. 1A; Goff et al., 2020; Terry et al., 2017).

    To take proper precautions against tsunami, numerous forward hydrodynamic simulations have been generated that
illustrate the worst-case scenarios of tsunami wave generation, propagation, and inundation as a result of the geological sources in the area (e.g., Qiu et al., 2019; Li et al., 2019; Li et al., 2016; Li et al., 2015; Okal et al., 2011; Wu and Huang, 2009; Megawati et al., 2009; Liu et al., 2022). However, the mitigation value of these simulations has not been assessed due to insufficient constraints from modern observation, historical accounts, and geological evidence. The modern tsunamis that have originated in the South China Sea have been insignificant, including those that followed the 1994 and 2006 earthquakes, with
wave heights reaching less than 60 cm (Fig. 1A; Central Weather Bureau, 2022). Historical accounts of events such as the 1604, 1640, and 1661 earthquakes are usually vague and only describe seawater overflow, which is not necessarily related to tsunami (Lau et al., 2010; Keimatsu, 1963; Nakamura, 1935; Lei and Ou, 1991). Accounts of the late 18th century tsunami in SW Taiwan remain controversial in terms of several aspects including age (1781 or 1782), death toll (one or tens of thousand), the extent of inundation (tens of meters or tens of kilometers), and whether the event was coupled with an earthquake (Li et
al., 2015; Lau et al., 2010; Liu et al., 2022). Recent geological investigations have identified three event layers from the 4th to the 16th centuries across the South China Sea (Lu et al., 2019; Sun et al., 2013; Ramos et al., 2017; Yang et al., 2019). However, further information is required for an accurate quantitative evaluation of the landward inundation, wave heights, and run-ups.

    In this study, the Penghu Islands in the Taiwan Strait in the NE South China Sea were selected for further geological investigation because of their close proximity to several tsunami sources and the previous discovery of three paleotsunami
deposits in the area (Fig. 1A; Lu et al., 2019), with a new and younger marine gravel deposit from the 17th century identified at several coastal localities. The cliff-top basalt boulders at one of the coastal exposure sites provide crucial clues for estimating wave heights and periods during sediment transport and deposition, and have been considered in this study for the first time. The matrix-supported boulders and the fine-grained matrix are well preserved and outcropped, leading to a unique situation in which key facies can be observed and used to constrain both transport modes and flow processes.

To better distinguish between tsunami and typhoon wave processes in the gravel deposit and reduce uncertainties in the wave estimation, multiple previously modified incipient motion formulas were utilized with the constraints obtained from the facies, stratigraphy, geomorphology, and modern observations (Nandasena, 2020; Cox et al., 2020; Kennedy et al., 2021; Barbano et al., 2010; Lorang, 2011; Nandasena et al., 2011; Pignatelli et al., 2009; Nott, 2003; Nandasena et al., 2022). Modifications and constraints include the virtual boulder dimensions, maximum lifting surface, sediment sources, transport
distances, shore slope angle, bathymetry, ancient sea level, modern records of typhoon waves, surges, and tsunamis, and estimated heights of 100 year significant waves and surges. Radiocarbon dating facilitates accurate determination of the



depositional age of the boulders and allows correlation with probable historical typhoon and tsunami events. The study is aimed at presenting accurate estimations of wave extent and boulder transport during impact and improving our understanding of tsunami risks by comparing the results with those obtained in previous simulations.

**Figure 1: Geomorphology of the study area. (a) Regional map of Taiwan with earthquakes and tsunamis from the 17th century (Keimatsu, 1963; Central Weather Bureau, 2022). (b) Map of Penghu Islands with outcrop localities. X–X' transect manifests a minor bathymetric gradient of 20:8380.**





## 2 Study area

The Taiwan Strait, is a narrow and shallow channel that lies northwest of Taiwan, connecting the East and South China Seas from NE to SW, with an average width of 180 km and a dominant water depth of 20–80 m (Fig. 1a). The Penghu islands in the southern strait are composed of more than 90 units of Miocene basalt platform and are mainly 10–20 m above sea level (a.s.l.), fringed by rocky shore terraces and platforms that reach a depth of 20 m offshore (Fig. 1B; Yen and Lee, 2017). Although Penghu is not far from the active plate convergence at the Taiwan orogen, the region is tectonically stable and is often used as a fixed point for geodetic reference (Yu et al., 1997). The sea level has been falling at approximately 5.1 cm per century from the Late Holocene in the region, decreasing by 2.4 m from a maximum 4700 years ago to that observed in the present (Chen and Liu, 1996).

Both Taiwan and the Taiwan Strait are regularly subjected to powerful typhoons and several wave height and period records have been set at Paisha in Penghu Islands and Tainan, Kaohsiung, Elaunbi, and Hualien in southern–eastern Taiwan (Table 1, Fig. 1A–B; Central Weather Bureau, 2022). The estimated significant wave heights over 50- and 100-year periods are available from the Paisha, Tainan, Kaohsiung, and Hualien Buoys (Jiang et al., 2012, 2011; Chang, 2009). In general, typhoon waves are less intense in the narrow, shallow, and tide-dominated Taiwan Strait, as shown by the wave height maximum at Paisha (6.8 m) as compared to the 11.9–17.8 m maxima observed at the other buoys that are adjacent to deep, open water.

Records of modern and ancient typhoon surges from the 17th century are available for the Penghu area with a mesotidal range of 2.0–3.2 m (Hsu, 2007; Central Weather Bureau, 2022). A modern surge maximum of 1.8 m a.s.l. resulted from Typhoon Mitag in 2019. From 1997 to 2021, 87 % of the 118 observed surges were 0.8–1.5 m a.s.l. with a mode of 1.1 m a.s.l. Ancient records include a 1.2 m a.s.l. surge in 1683 and two surges that reached 1.5 m a.s.l in 1832. A modern surge maximum of 1.8 m a.s.l. is tentatively inferred to as the 100 year surge in this study.

Records of wave events at the Penghu Islands start in the thirteenth century, and no ancient tsunamis were recorded. The modern earthquake-related tsunami in 1994 sets the wave height record at 0.4 m (Fig. 1A; Central Weather Bureau, 2022).



**Table 1.** Yearly and monthly wave records and estimated significant wave heights at buoys in the study area. Buoy locations can be seen in Fig. 1a.

| Time | Observation | Max significant wave[1] | | | | Probable significant H | |
|---|---|---|---|---|---|---|---|
| | | | | | | Period | |
| | | H | P | D | Date | 50 years | 100 years |
| Paisha (Penghu Islands; 2006–2021) | | | | | | | |
| Aug. | 9554 | 6.8 | 8.7 | 337 | 2015/08/08[2] | 7.3[6] | |
| Oct./year | 10821 | 6.8 | 10.4 | 45 | 2011/10/03[3] | | |
| Tainan (Qigu; 2006–2021) | | | | | | | |
| Aug./year | 10594 | 14.0 | 10.4 | 258 | 2015/08/08 | 11.6[7] | 12.8[7] |
| Kaohsiung (Mituo; 2012–2021) | | | | | | | |
| Aug. | 6237 | 10.9 | 17.0 | 281 | 2015/08/08 | 10.0[8] | 10.8[8] |
| Sep./year | 6662 | 12.2 | 23.2 | 236 | 2016/09/14[4] | | |
| Elaunbi (2002–2021) | | | | | | | |
| Aug. | 12660 | 10.7 | 13.1 | — | 2009/08/08[5] | | |
| Sep./year | 11868 | 17.8 | 13.1 | 168 | 2016/09/14 | | |
| Hualien (2002–2021) | | | | | | | |
| Aug./year | 12420 | 11.9 | 13.1 | 90 | 2015/08/08 | 15.7[8] | 16.3[8] |

H: wave height (m), P: wave period (s), D: wave direction (degrees).
[1]Wave statistics from buoys at a depth of 20–40 m (Central Weather Bureau, 2022), [2]Typhoon Soudelor, [3]Atmospheric pressure low, [4]Super typhoon Meranti, [5]Typhoon Morakot, [6]Chang (2009), [7]Jiang et al. (2012), [8]Jiang et al. (2011).

## 3 Materials and methods

### 3.1 Cliff-top boulder and incorporative hydrodynamic approach

The studied cliff-top basalt boulder is found in outcrop CT-1, which lies on the south shore of a narrow and shallow interisland channel, denoted the CT Channel in Fig. 2a. The channel is 350–500 m wide and largely 2 m deep (Fig. 2b–c).

The largest boulder unit, hereafter denoted CTB (CT boulder), was selected for calculation because the other boulders in the outcrop are significantly smaller (Fig. 3a). The cliff-top boulders are supported by a gravel and mud matrix that forms a



lateral gravel layer (MECT-1) that pinches out from 2.5 to 4.0 m a.s.l. Marine shells and rounded pumice pebbles that are abundant in both matrix and gravel layer, and are also found on modern beaches in the region (Fig. 3b), are absent in the underlying basalt basement, basal soil, and overlying angular-gravel colluvium. Accordingly, a marine sediment origin for the
cliff-top boulder and gravel layer is assumed.

     The CTB metrics: 2.91 g cm$^{-3}$ density, 0.73 m$^3$ volume, 2.10–2.15 t mass, an axis length of 1.71–0.69 m, and a projected surface of 1.66–1.01 m$^2$, were measured using Autodesk Meshmixer and Context Capture (Fig. 3c). The CTB is angular in shape, indicating a joint-bounded source in the intertidal–subtidal zone because subaerial spheroidal weathering and rock residue are well developed all over the islands, including the CT-1 basalt cliff (Fig. 3a). An intertidal rock exposure that is
located 0.5 m below sea level and comprises isolated and stacked boulders of sizes and shapes that are comparable to the CTB may be the source of the studied boulder (Fig. 3d). This rock exposure forms a wave-cut bench and an inflection point on the shore profile and indicates a transport distance of 31.7 m to the CT-1 outcrop (Fig. 2; Lorang, 2011). A transect through the outcrop to the probable source and deepest channel shows slope angle changes from at 21.5$^o$ at the supratidal to 3.6$^o$ at the beach–intertidal and 0.01$^o$ at the subtidal (YY' in Fig. 2c).

The incorporative hydrodynamic approach is performed in two parts: an initial estimation of the minimum wave heights and periods required to initiate transport of the CTB using the modified incipient motion formulas, and distinguishing between the typhoon and tsunami wave origins by comparing the estimated results with modern and ancient records of waves associated with tsunamis, typhoons, and the 100 year significant wave and surge heights. Comparing estimation with records integrates the constraints associated with the geomorphology, bathymetry, ancient sea level, and the ratio of wave height to water depth
at the wave break (Cox et al., 2020).

     The previously modified Nott equations that use the maximum lifting surface, shore slope angle, and transport modes of sliding, rolling, and saltation into account and cover submerged, subaerial, and joint-bounded scenarios were adopted in the initial calculation (Table 2; Nott, 2003; Pignatelli et al., 2009; Barbano et al., 2010; Lorang, 2011; Nandasena et al., 2022). The modifications also incorporate the virtual boulder shape, volume, and projected surfaces into the drag and lift coefficients
(Table 2; Nandasena et al., 2022). The Hudson formula is then adopted for independent estimates of the beach–intertidal zone (Lorang, 2011; Hudson, 1953). A ratio of 0.25 was obtained for tsunami and storm wave heights using 1.0 and 2.0 as Froude Numbers to describe the storm and tsunami waves, respectively. The wave periods are then derived from (1) the estimated heights, and (2) the modified Noortmets equation for maximum water flooding (Noormets et al., 2004; Barbano et al., 2010; Lorang, 2011). For the Noortmets equation, the cliff-top elevation was measured at the outcrop, 2.5 m a.s.l., and the minimum
run-up height was marked by the MECT-1 pinch-out 4.0 m a.s.l. (Fig. 3a). The regional bathymetric gradient, which is used to delineate the ratio of wave height to water depth at the wave break in the second part of the approach, was derived from the XX' transect between the CT channel and an abandoned offshore buoy in the NE (Fig. 1B, Table 2; Collins, 1970; Noormets et al., 2004). The water depth change is 20 m, the distance is 8380 m, and the ratio is 0.73.

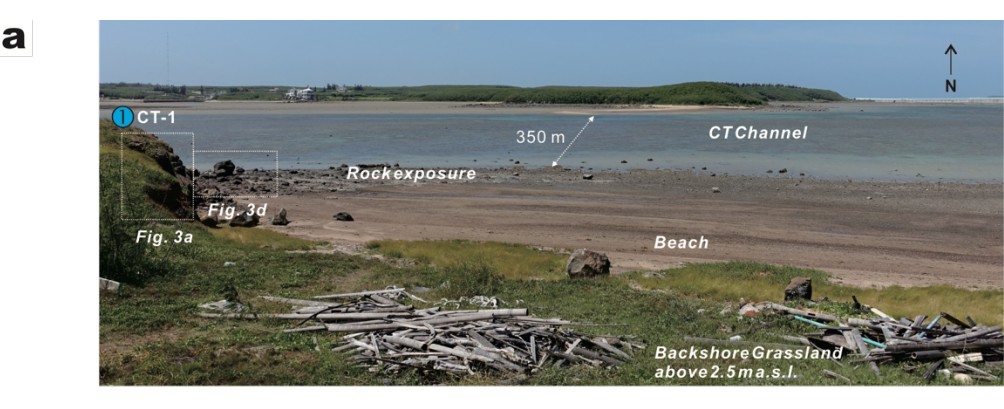

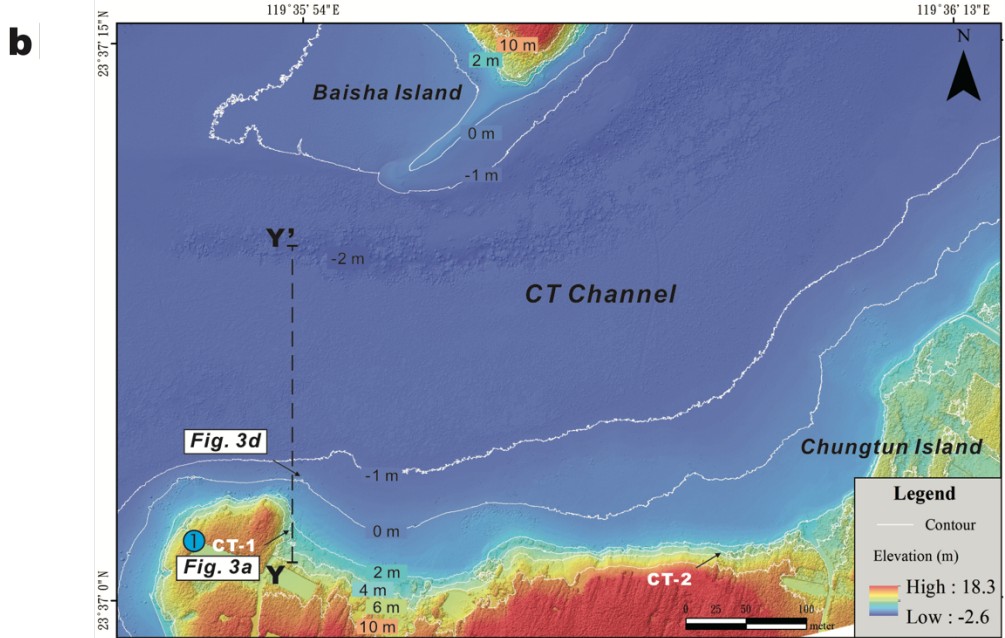

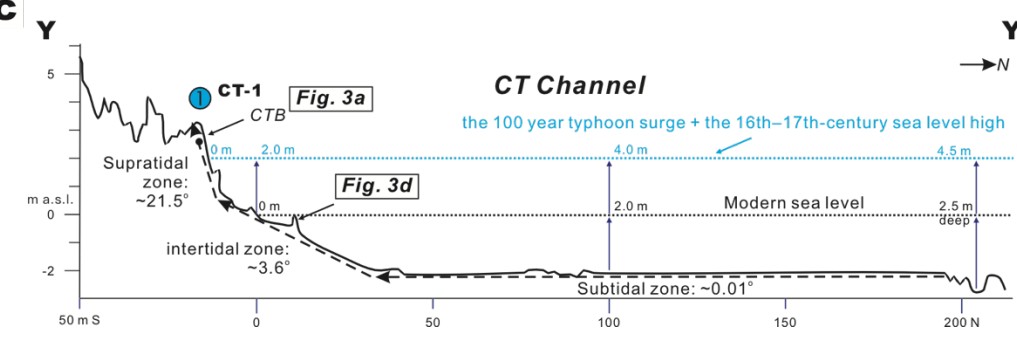

Figure 2: Regional view and bathymetry of outcrop CT-1. Location is illustrated in Fig. 1b (blue circle 1). (a) CT-1 shore and interisland CT Channel at low tide. Channel is less than 2.5 m in depth. (b) DEM image, and (c) YY' transect showing an increase in the slope angle from the flat subtidal to the gentle intertidal and steep supratidal zones. The blue dotted line in the transect marks the estimated maximum depth in the 17th century when the sea level was 0.2 m higher (Chen and Liu, 1996) and is superimposed by the 100 year typhoon surge at 1.8 m a.s.l.





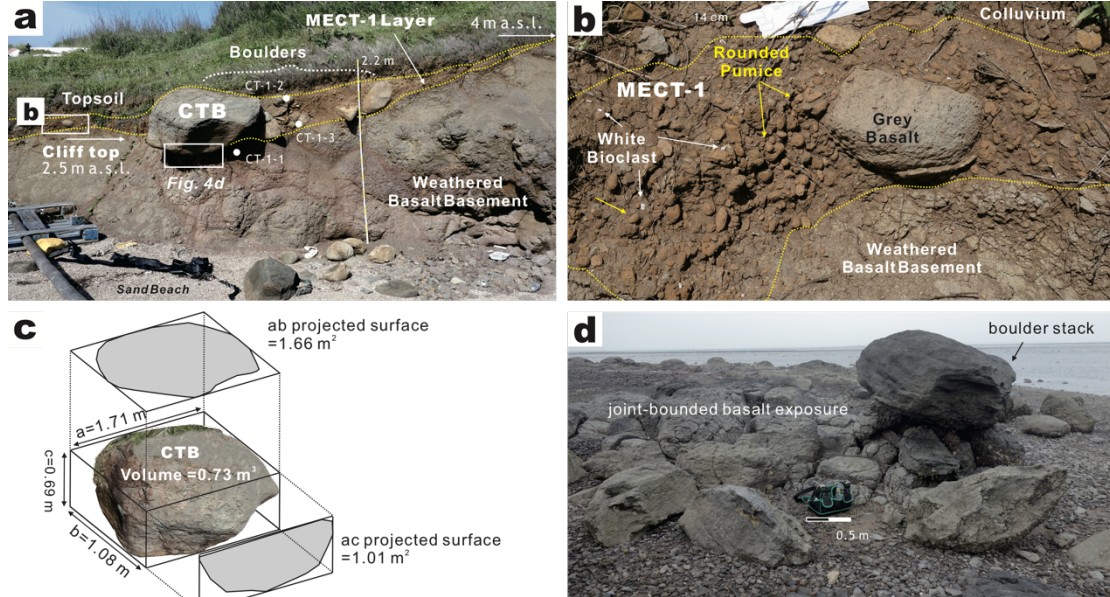

**Figure 3: Outcrop views and scanned image of the CTB, MECT-1 gravel, and exposed intertidal rock. Location is provided in Fig. 2. (a) CT-1 section from the basalt basement cliff up to the MECT-1 layer and topsoil. Matrix-supported boulders, including the CTB, are prominent in the MECT-1 layer that pinches out from 2.5 to 4.0 m a.s.l. Note the spheroidal weathering and rock residues**

**in the basement. CT-1-1–3 mark the radiocarbon sample locations. (b) Close-up view of the matrix-supported MECT-1 that is composed of rounded brownish pumice, grey basalt, and white bioclast. (c) CTB metrics obtained using Autodesk Meshmixer and Context Capture. (d) Intertidal joint-bounded basalt exposure, boulder stacks, and isolated boulders. Note that boulders are comparable to the CTB in both size and shape.**

**Table 2. Modified incipient motion formulas adopted in this study.**

| Initial transport mode | Submerged | Subaerial | Joint bounded |
|---|---|---|---|
| Sliding* | $H \geq \dfrac{2C_{vv}\left(\frac{\rho_s}{\rho_w}-1\right)c(\mu_s\cos\theta+\sin\theta)}{C_{dv}\left(\frac{c^2}{b^2}\right)+C_{lv}}$ | | — |
| Rolling/overturning* | $H \geq \dfrac{2C_{vv}\left(\frac{\rho_s}{\rho_w}-1\right)c\left(\cos\theta+\frac{c}{b}\cdot\sin\theta\right)}{C_{dv}\left(\frac{c^2}{b^2}\right)+C_{lv}}$ | | — |
| Saltation/lifting* | $H \geq \dfrac{2C_{vv}\left(\frac{\rho_s}{\rho_w}-1\right)c\cdot\cos\theta}{C_{lv}}$ | | $H \geq \dfrac{2C_{vv}\left(\frac{\rho_s}{\rho_w}-1\right)c(\cos\theta+\mu_s\sin\theta)}{C_{lv}}$ |
| Hudson Formula** | | $H=\left(\left(K_D\left(\frac{\rho_s-\rho_w}{\rho_w}\right)^3\cot\alpha\right)V\right)^{1/3}$ | |
| Wave period I** | | $P=\left(\frac{2}{g}\right)\cdot\left(\frac{\rho_w}{\rho_s-\rho_w}\right)\left(\frac{C_{dv}}{\tan\theta}\right)\left(\frac{h_{clast}}{b}\right)\sqrt{g\cdot(W_b+H_S)}$ | |



| II*** | $P = \dfrac{5 \cdot X_{max}}{\cos\delta \cdot \sqrt{g(R-E)}}$ |
|---|---|
| Wave height/water depth ratio**** | $H_S/W_b = 0.72 + 5.6 \cdot \tan\beta$ |

\* Nandasena et al. (2022); \*\* Lorang (2011); \*\*\* Barbano et al. (2010); \*\*\*\* Collins (1970).

Wave height: $H$; Storm wave weight: $H_S$; Wave period: $P$; Water depth at wave break: $W_b$.

CTB axis lengths: a = 1.71 m; b = 1.08 m; c = 0.69 m. CTB volume: $V$, 0.73 m³. Projected ab surface: $A_{ab}$, 1.66 m²; Projected ac surface: $A_{ac}$, 1.01 m². CTB density: $\rho_s$, 2910 kg m⁻³; Seawater density: $\rho_w$, 1020 kg m⁻³.

Virtual lift coefficient: $C_{lv} = \frac{A_{ab}}{ab} C_l$; Virtual drag coefficient: $C_{dv} = \frac{A_{ac}}{ac} C_d$; Virtual volume coefficient: $C_{vv} = \frac{V}{abc}$.

Lift coefficient: $C_l$, 0.178; Drag coefficient: $C_d$, 1.95; Static friction coefficient: $\mu_s$, 0.7; $K_D$ for smooth stone: 1.5; Gravity: $g$, 9.8 m s⁻².

Shore slope: $\theta$; Beach slope: $\alpha$; Regional slope: $\beta$, 20/83800 (X-X' in Fig. 1b); Mean CT-1 coastal overland slope: δ, 21.5°.

CTB elevation: $h_{clast}$, 2.5 m a.s.l.; Maximum flooding distance from the probable intertidal source to CT-1 outcrop: $X_{max}$,

31.7 m; Run-up of event layer: $R$, 4.0 m a.s.l.; Cliff top elevation: $E$, 2.5 m a.s.l.

**3.2 Coastal outcrops and facies-stratigraphic analysis**

To better obtain the facies and stratigraphic constraints on the transport and deposition of the CTB, CT-1 and three additional outcrops with supratidal gravel layers that were deposited during a marine event were investigated; the adjacent

outcrop CT-2 (Figs. 2b and 4a) and the two distant outcrops WT and STS (blue circles 2 and 3 in Fig. 1b; Fig. 4b–c). A previously reported Suogang/SG outcrop was included for comparison (blue circle 4 in Fig. 1B; Lu et al., 2019). All studied outcrops are in the supratidal zone with top surfaces at an elevation of 2.5–5.3 m a.s.l.

Analyses of the sedimentary facies, granulometry, and radiocarbon dating were applied to the new outcrops (CT-1, 2, WT, and STS). Facies analysis integrates lithology with sedimentary structure and fossil content. Granulometric samples were

collected from sections CT-2 and STS, characterized by their large total thicknesses and massive marine gravel layer texture, and analyzed using a sieve shaker and a Microtrac S3500 laser particle size analyzer. Charred material samples from the marine gravel layers and ambient deposits were sent to Beta Analytic, Miami, Florida, USA, for radiocarbon dating. The modeled ages of both new and previously-reported marine gravel layers were calculated using OxCal 4.4 from the University of Oxford and the IntCal13 calibration curves (Ramsey, 2021; Reimer et al., 2013).

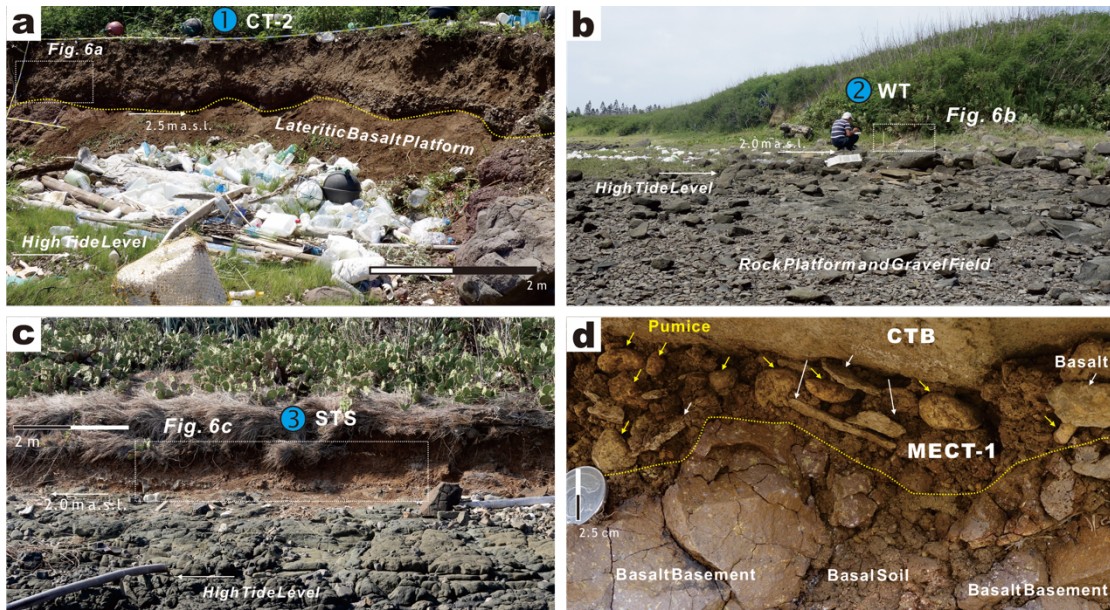

**Figure 4: Regional and close-up views of the investigated Holocene sections at low tide. (a) CT-2 section lying above the basalt platform at 2.5 m a.s.l. (location in Fig. 2b). (b) WT section lying above the basalt platform at 2.0 m a.s.l. (location in Fig. 1b). (c) STS section with the basalt platform top at 2.0 m a.s.l. (location in Fig. 1b). (d) Mud and gravel matrix beneath the CTB in MECT-1 layer (location in Fig. 3a). The gravel component includes rounded pumice (yellow arrows) and platy–angular basalt (white arrows).**

## 4 Results

### 4.1 Wave conditions during CTB transport and deposition

#### 4.1.1 Wave height

The estimated minimum storm wave height required to initiate CTB transport of 1.22–9.5 m (Table 3) is well within the 6.8–17.8 m height observed for modern typhoon waves and the estimates of the significant waves for the 50- and 100-year periods (Table 1; Central Weather Bureau, 2022). These results seem plausible because western Pacific typhoons are notoriously powerful and the CTB is relatively small in volume and mass.

However, taking the ancient bathymetry during deposition, facies, and probable transport modes into account renders typhoon waves unsuitable candidates for the deposition of CTB at its current cliff-top position. According to the dating results, the CTB was probably deposited in the 16th–17th centuries (CT-1-1 and 3 in Fig. 3a and Table 4). During this period, the maximum water depth in the CT Channel could increase from 2.5 to 4.5 m because the sea level was approximately 0.2 m higher than it is at present, allowing a 100 year surge of 1.8 m a.s.l. to occur (Fig. 2C; Chen and Liu, 1996; Central Weather Bureau, 2022). The tallest typhoon waves that can form in the subtidal–intertidal water depth (1.46–3.29 m; wave height = $0.73 \times$ water depth), can initiate sliding and rolling of the CTB upon breaking (Table 3). This sliding–rolling movement is



likely to cease in the steeper supratidal zone because the estimated minimum waves are too tall to form in the shallow depths of the inundated supratidal zone and will break earlier before attenuating rapidly and exponentially landwards (Barbano et al., 2010; Goto et al., 2009). The estimated minimum waves of 8.5–9.5 m that would be required for saltation and lift from the probable intertidal joint-bounded source cannot occur in the 4.5 m deep CT Channel.

The estimated minimum heights of the tsunami waves fall in two groups: (1) 0.33–0.51 m for initiation of sliding–rolling, 210 which is comparable to the 0.4 m high tsunami that inundated the Penghu Islands in 1994, and (2) 2.13–2.38 m for the saltation and joint-bounded scenario (Table. 3).

Based on the facies and CT-1 geomorphology, it appears that these minima are insufficient for deposition of the CTB at its present position. The CTB is floored by the pumice-bearing gravel and mud matrix above the cliff basement (Fig. 4d) and the gravel layer are matrix-supported (Fig. 3a–b). It indicates that the terminal CTB transport on the cliff top was controlled 215 by suspension-rich turbulent flow process and was very likely rolling or saltation before deposition. This is in agreement with the bore-like long-wave-length waves that tend to lead to boulder rolling and saltation (Imamura et al., 2008), especially for boulders that are smaller than 2 m in length, as is the CTB (Nandasena et al., 2011). The 0.44–0.51 m high tsunami waves that would be required to initiate rolling of the CTB would cause it to rotate and reach 2.14–2.21 m a.s.l., before finally depositing it on the slope of the 2.5-m-high cliff (Fig. 3a). The tsunami waves need to be 0.8 m high to rotate the CTB to reach the cliff 220 top elevation and to be over 1.65 m high (0.8 + 1.7/2) to rotate the CTB median point onto the cliff top. In the sliding–rolling transport, any accompanying coeval finer sediment would be deposited separately below.

The minimum tsunami wave height of the saltation and joint-bounded scenarios, 2.13–2.38 m, is sufficient to lift the 1.7 m long CTB to 3.83–4.09 m a.s.l. onto the cliff top while depositing the coeval finer sediment on the cliff slope below. However, further overtop wave run-up and sediment transport are required to prevent the CTB from falling back down the cliff as a result 225 of tsunami backwash (Pignatelli et al., 2009; Imamura et al., 2008; Lorang, 2011). Since the terminal CTB was probably moving in a rolling–saltation manner before deposition, the substantial depth required for the cliff-top run-up/overflow is unlikely to result from tsunami waves that are only 2.13–2.38 m in height (Figs. 3a and 4d).

Collectively, these results suggest that a tsunami wave height of 3.0 m would be required to maintain the overtop flow at the estimated minimum height of 0.51 m for the terminal CTB deposition on the 2.5 m high cliff top. In addition, the MECT- 230 1 pinch-out at 4.0 m a.s.l. marks the minimum run-up and local wave height at 4.0 m (Fig. 3A; Pignatelli et al., 2009; Nanayama and Shigeno, 2006; Paris et al., 2018).

### 4.1.2 Wave period

The estimated wave period ranges from 3.4–15.0 s in the supratidal zone to 19.6–132.9 s in the intertidal zone and $6.47 \times 10^3$–$3.55 \times 10^4$ s in the subtidal zone (Table 3). According to the formulae used (Table 2), the prominent seaward increase 235 is primarily controlled by the slope angle, which decreases drastically to 0.01° in the nearly horizontal subtidal zone (Fig. 2; Barbano et al., 2010; Lorang, 2011). The observed increases in water depth and wave height also favor a wave period increase.





Compared to the wave records from Taiwan, the typhoon waves appear irrelevant in terms of the estimated wave period (Table 1). Only the estimate of 6.8–15.0 s that was obtained for the supratidal zone is comparable to the modern wave maximum of 8.7–23.2 s, which occurs offshore at depths of 20–40 m (Central Weather Bureau, 2022).

In contrast, the estimated subtidal period that is associated with the minimum tsunami wave height of 1.8–4.9 h ($6.47\times10^3$– $1.77\times10^4$ s) is within the range typical of modern tsunamis (Barbano et al., 2010; Lorang, 2011). The estimated period extends to 5.6 h when the waves are 3.0 m high, allowing waves to overflow the 2.5 m high cliff top and maintain the minimum 0.5 m depth. The successive shortening of the estimated period in the intertidal–supratidal zone probably responds to the shoaling that is due to deceleration of the tsunami wave and also causes a landward decrease in wavelength alongside an increase in the

wave height. Therefore, the CTB transport is inferred to be associated with long-periodicity tsunami or tsunami-like composite waves produced by interplay between typhoon waves, surge, infra-gravity waves, and astronomical high tides (Baumann et al., 2017; Brill et al., 2016; Nakamura et al., 2014; Sohn and Sohn, 2019; Watanabe et al., 2017)


**Table 3. Results of wave estimation from equations in Table 2.**

| Wd†, $\theta$ | | Supratidal 2.0 m>Wd, 21.5º | Intertidal 2.0<Wd<4.0 m, 3.6º | Subtidal 4.0<Wd<4.5 m, 0.01º |
|---|---|---|---|---|
| **Sliding** | $H_S, P_S, W_b$ | 1.78, 6.8, 2.4 | 1.32, 39.2, 1.8 | 1.22, $1.29\times10^4$, 1.8 |
| | $H_T, P_T$ | 0.44, 3.4 | 0.33, 19.6 | 0.33, $6.47\times10^3$ |
| **Overturning/ Rolling** | $H_S, P_S, W_b$ | 2.03, 7.3, 2.8 | 1.80, 45.6, 2.5 | 1.74, $1.55\times10^4$, 2.4 |
| | $H_T, P_T$ | 0.51, 3.6 | 0.45, 22.9 | 0.44, $7.73\times10^3$ |
| **Saltation** | $H_S, P_S, W_b$ | 8.50, 15.0, 11.6 | 9.14, 103.2, 12.5 | 9.16, $3.55\times10^4$, 12.5 |
| | $H_T, P_T$ | 2.13, 8.5 | 2.28, 52.7 | 2.29, $1.77\times10^4$ |
| **Joint bounded** | $H_S, P_S, W_b$ | | 9.5, 132.9, 13.0 | |
| | $H_T, P_T$ | | 2.38, 52.7 | |
| **Hudson F.** | $H_S, P_S, W_b$ | | 4.88, 95.1, 6.68 | |
| **Ps/Noormets Eq.** | 44.4s | | | |

†: maximum water depth during the 100 year surge in the 17th century; $\theta$: shore slope.

$H_S$: storm wave weight (m); $P_S$: storm wave period (s); $W_b$: wave break depth (m), i.e., $H_S 0.73^{-1}$.

$H_T$: tsunami wave weight, i.e., $0.25 \cdot H_S$ (m); $P_T$: tsunami wave period (s).




**Table 4.** AMS Radiocarbon dating results. See locations in Figs. 3a and 5a–b.

| Sample locations | Anal. No. | Material | δ¹³C (‰) | Conventional age | 2 Sigma calibration |
|---|---|---|---|---|---|
| CT-1-1 | 528841 | Charred material | $-13.5$ | $280 \pm 30$ BP | Cal 1500 to 1658 CE |
| CT-1-3 | 539055 | Charred material | $-16.1$ | $250 \pm 30$ BP | Cal 1510 to 1667 CE |
| CT-1-2 | 528843 | Charred material | $-17.1$ | $105.76 \pm 0.39$ pMC | Cal 1957 to 2009 CE |
| CT-2-1 | 539056 | Charred material | $-16.8$ | $540 \pm 30$ BP | Cal 1326 to 1439 CE |
| CT-2-2 | 538845 | Charred material | $-18.0$ | $450 \pm 30$ BP | Cal 1418 to 1474 CE |
| WT-1 | 563788 | Charred material | $-13.5$ | $420 \pm 30$ BP | Cal 1455 to 1625 CE |
| WT-2 | 563789 | Charred material | $-17.0$ | $200 \pm 30$ BP | Cal 1645 to 1948 CE |
| STS-E-1-2 | 510514 | Charred material | $-12.4$ | $110 \pm 30$ BP | Cal 1681 to 1939 CE |
| STS-E-2 | 484573 | Charred material | $-12.2$ | $320 \pm 30$ BP | Cal 1487 to 1640 CE |
| STS-E-3 | 484574 | Charred material | $-13.0$ | $350 \pm 30$ BP | Cal 1479 to 1630 CE |
| STS-E-4 | 484575 | Charred material | $-12.9$ | $390 \pm 30$ BP | Cal 1437 to 1506 CE |
| SG-3‡ | 435211 | Charred material | $-14.6$ | $890 \pm 30$ BP | Cal 1040 to 1220 CE |
| SG-2‡ | 405582 | Foraminifers | $+0.2$ | $960 \pm 30$ BP | Cal 1030 to 1230 CE |
| SG-4‡ | 405583 | Charred material | $-13.7$ | $650 \pm 30$ BP | Cal 1291 to 1403 CE |
| SG-5‡ | 438347 | Charred material | $-15.7$ | $100.6 \pm 0.3$ pMC | post 1950 CE |

‡ Lu et al. (2019).

## 4.2 Supratidal Upper Holocene facies


### 4.2.1 Description

The newly- and previously-studied outcrops sit atop the supratidal shore platforms at 2.5 m a.s.l. (CT-1 and 2; Figs. 3a and 4a) and 2.0 m a.s.l. (STS, WT, and SG; Fig. 4a–c). The Upper Holocene deposits are 0.4–1.2 m thick, mainly composed of common colluvium, and form dune deposits on the islands (Fig. 5A–B; Yen and Lee, 2018; Lu et al., 2019). Overlying the

weathered, lateritic basalt basement, the Upper Holocene is composed of the mottled/weathered gravelly colluvium (CT-1 and 2 sections; Figs. 3b and 6a), bioclastic sand (WT section; Fig. 6b), and brownish sandy soil (STS and SG; Fig. 6c–d) and the top humic soil. The mottled brownish sandy soil and colluvium are poorly sorted (CT-2-03, STS-01, 03, 04, and 06 in Fig. 5c)


and contain angular basalt debris (Figs. 3a–b, 4d, 6a, c–d). The bioclastic sands are better sorted and dominated by shell fragments and foraminifers, forming local dunes (Figs. 4b and 6b). According to the radiocarbon dating results, the newly-

studied sections have accumulated since the 15th century while section SG started to accumulate in the 10th century (Fig. 5A–B, Table 4; Lu et al., 2019).

The gravel layers that are associated with a supratidal marine event (ME) are 10–70 cm thick and lie at an elevation of 2.0–4.5 m a.s.l. (MECT-1 and other ME layers in Fig. 5a–b). The ME layers are poorly sorted, matrix-supported, massive/structureless, and in sharp basal contact with the weathered basalt residues and mottled deposits (Figs. 3b and 6). The

sharp contact is evidenced by abrupt upward increases in granulometry from the lateritic basalt basement to the CT-2-01 sample, from STS-01 to 02, and from STS-04 to 05 (Fig. 5b–c). The gravel compositions include basalt, pumice, and bioclast, with the bioclast dominating in the MEWT, MESTS-a, and b layers (Figs. 3a and 6; Fig. 5b; Lu et al., 2019). The basalt grain size is pebble–cobble, increasing to small–medium boulders in outcrop CT-1 (Fig. 3a). The pumice is rounded to subrounded in shape, pebble sized, and rusty–yellowish brown in color (Figs. 3b and 6a–b). The bioclasts include coral, mollusks, and

foraminifera, and are characterized by articulated bivalves and well-preserved gastropods in the MEWT, MESTS-a, and b layers (Fig. 6b and d). The matrix contains sediments that are similar to the underlying deposits and basement soils, and includes weathered–lateritic angular basalt debris, mottled brownish sandy soil, and bioclastic sand (Figs. 4d, and 6).


**Figure 5: Stratigraphic correlation between outcrops and granulometry in this study. (a, b) Stratigraphic columns and correlation. Radiocarbon dating details can be found in Table 4. (c) Granulometry of the STS and WT outcrops showing an abrupt grainsize increase from the underlying samples to the ME deposit samples (CT-2-01 and 02, STS-02 and 05).**
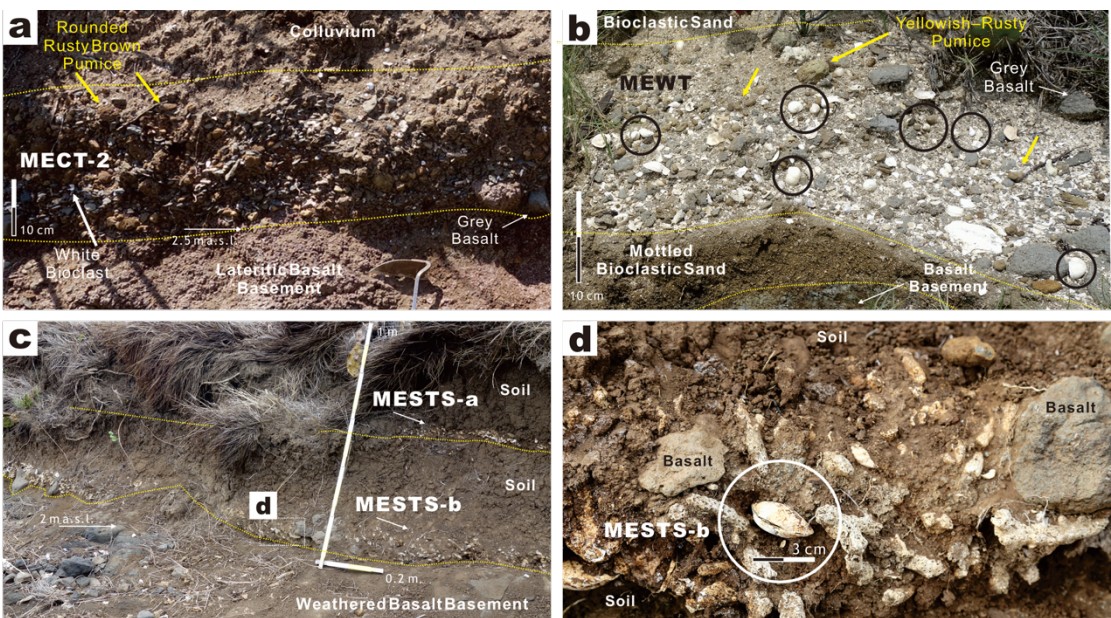

**Figure 6: Outcrop views of the marine event gravel layers in this study. See locations in Fig. 4. (a) Matrix-supported MECT-2 layer characterized by pebbles of rounded rusty brown pumice, grey basalt, and white bioclast. (b) MEWT layer characterized by well-preserved gastropods (black circles) and dominated by white bioclast with grey basalt and yellowish–rusty grey rounded pumice. (c) STS section showing intercalated soil and marine event gravel (MESTS-a and b). (d) Close-up view of articulated bivalves (white circle) among the coral and basalt debris in the matrix-supported MESTS-b layer.**

### 4.2.2 Interpretation

The mottled colluvium and dune deposits are interpreted as terrestrial and/or supratidal accumulations that have been subjected to post-depositional soil formation. The soil preservation is attributed to the regional tectonic and geomorphological stability and the lack of wave and tidal reworking in the supratidal zone (Fig. 4a–c). The angular basalt colluvium in the CT-1 and 2 outcrops appears to be the result of rockslides from the relatively steep slopes (Figs. 2b and 4a). The mottled sandy soil (STS and SG outcrops) and bioclastic sand (WT outcrop) are likely to be supratidal dune deposits (Fig. 4b–c).

The sharp-based, poorly sorted, matrix-supported gravel layers are interpreted as event deposits that record initial scouring and rapid fallout from suspension-rich turbulent flows (Fujiwara and Kamataki, 2008; Lowe, 1982; Nanayama and Shigeno, 2006; Paris et al., 2018). Based on the coarse-grain size of cobble–boulders and the 1.7 m long CTB, the flow velocity, density, and turbulence were high; thus, the combined hydraulic lift and dispersive pressure could prevent settling during transport. The rapid fallouts are highlighted by the matrix-supported gravel fabric and the mixed particles of wide density and size range found in the vesicular pumice to carbonate bioclast and mafic igneous basalt (CT-2-01, 02, STS-02, and 05 in Fig. 5c).

The sediments of the gravel layers are coastal marine in origin and were reworked from the beach–dune environments to the high supratidal location. The beach-derived sediments include mixed bioclasts comprising coral, foraminifer, and mollusk





with pumice, all of which are commonly observed on local beaches (Figs. 3a and 4a–c). Modern beach pumice includes stranded rafts from the 2021 submarine eruption in the southern Izu-Bonin Arc, whereas pumice is absent in the Miocene
basalt and sedimentary basement (Yen and Lee, 2017; Yu et al., 2022). Ancient strands of pumice are commonly incorporated in backshore event redepositions and are reported from the past 2.4 ka in northern Taiwan (Yu et al., 2022). The redeposited supratidal material is represented by angular basalt, brownish soil, and bioclastic sand in the matrix and is present in the sections below the ME layers (Figs. 4d and 6a–d).

## 4.3 Age dating and correlation

The investigated Upper Holocene stratigraphy is best manifested by STS section because of the continuous exposure, the two ME gravel occurrences, and the radiocarbon dates in ascending and decreasing order (Fig. 5a–b). Layer MESTS-b is intercalated into sandy soil dating from the 15th–early 17th centuries, with the ambient soil layers of the MESTS-a dating from the 16th–17th centuries and the 17th–19th centuries. Laterally, the MESTS-b layer is comparable in age to the MECT-2 (CT-2 outcrop) which was deposited in the 14th–15th centuries at the earliest based on the age of the detrital sediment matrix. The
MESTS-b layer is also comparable in age to MESG-a (SG outcrop), which overlies soil from the 13th–14th centuries. MESTS-a is correlated with MECT-1 and CTB (CT-1 outcrop), which were deposited in the 16th–17th centuries, according to the dating results obtained for MECT-1 and the underlying soil. MESTS-a is also correlated with MEWT (WT outcrop), which lies between deposits from the 15th–16th centuries and the 17th–19th centuries.

The dating results are refined by the modeled ages from the calculation results obtained using OxCal 4.4 (Fig. 7, Table 5;
Ramsey, 2021). The modeled MESTS-a and b ages of 1444–1573 and 1524–1706, respectively, form a continuous time span with the 1575 midpoint. A modeled interval of 12–227 years lies between the layers.

Collectively, the CTB and the MECT-1 deposits are probably from the event that led to marine inundation in the 17th century (1575–1706, 1640±66). This event is reported for the first time on Penghu Islands and in the South China Sea region. The MECT-2 layer in the CT-2 outcrop records an earlier event that probably occurred in the late 15th–late 16th centuries
(1444–1575, 1510±65). The CT-2 deposit was reported upon previously, namely as layer MESG-a in the SG outcrop (Lu et al., 2019). The coeval deposits in the STS and CT-2 outcrops that were firstly examined in this study appear to constrain the northern extent of this marine inundation onto the Penghu Islands (Fig. 1b).




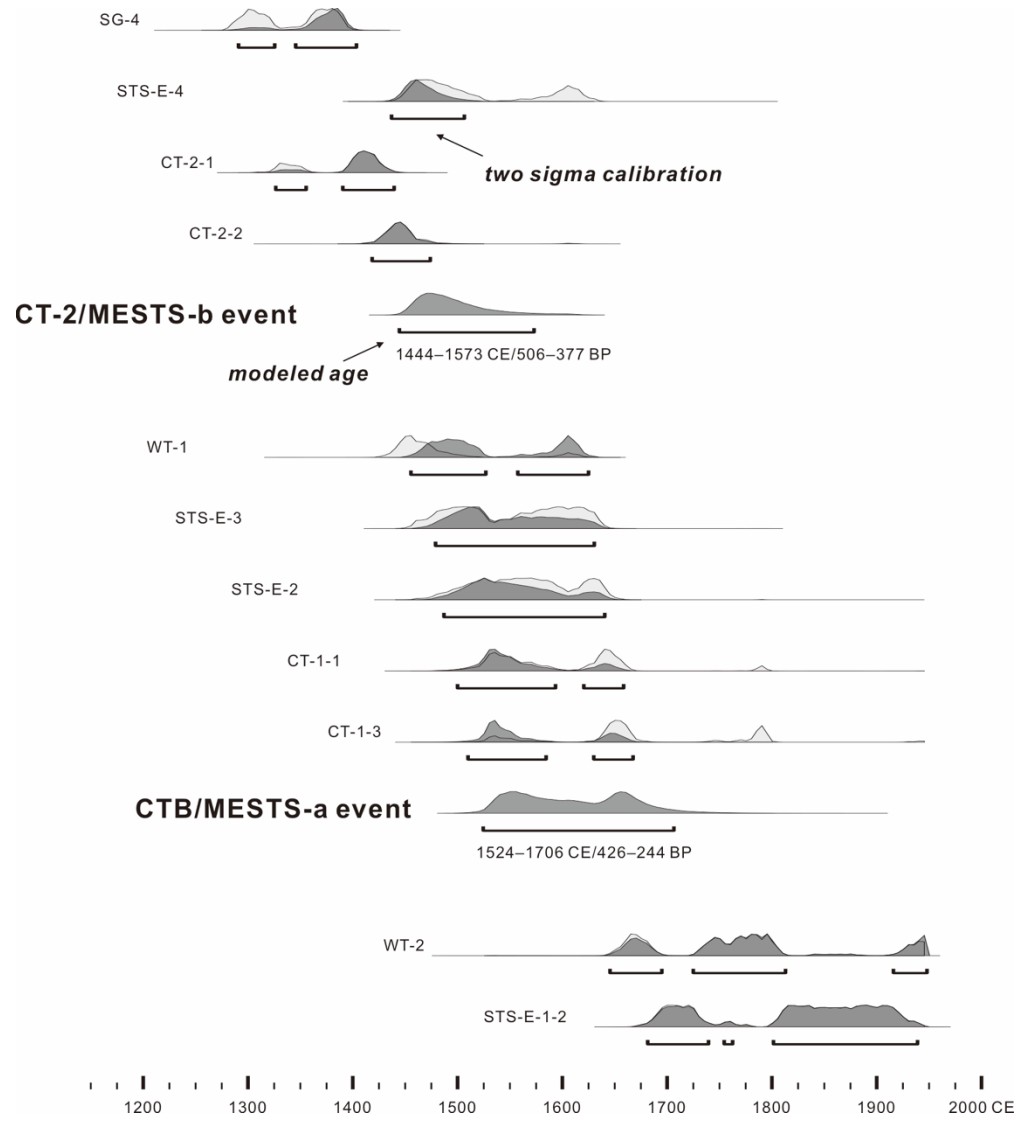

**Figure 7: Modeled ages of the radiocarbon samples and the CTB/MESTS-a and CT-2/MESTS-b layers obtained using OxCal v 4.4**
**(Ramsey, 2021). Further details are given in Table 5.**

**Table 5. Calculation results of OxCal v. 4.4 from Ramsey (2021) as compared to the marine event layers and radiocarbon samples in this study.**

| Gravel layer | Unmodeled ages (AD) | | | Modeled ages (AD) | | |
|---|---|---|---|---|---|---|
| and sample | from | to | % | from | to | % |
| SG | 1281 | 1395 | 95.4 | 1291 | 1403 | 95.4 |
| STS-E-4 | 1442 | 1631 | 95.4 | 1437 | 1506 | 95.4 |
| CT-2-1 | 1322 | 1437 | 95.4 | 1326 | 1439 | 95.4 |




| | | | | | | |
|---|---|---|---|---|---|---|
| CT-2-2 | 1413 | 1480 | 95.4 | 1418 | 1474 | 95.4 |
| MESTS-b | | | | 1444 | 1573 | 95.4 |
| WT-1 | 1426 | 1620 | 95.4 | 1455 | 1625 | 95.4 |
| STS-E-3 | 1461 | 1636 | 95.4 | 1479 | 1630 | 95.4 |
| STS-E-2 | 1484 | 1644 | 95.4 | 1487 | 1640 | 95.4 |
| CT-1-1 | 1505 | 1795 | 95.4 | 1500 | 1658 | 95.4 |
| MESTS-a | | | | 1524 | 1706 | 95.4 |
| WT-2 | 1643 | … | 95.4 | 1645 | 1948 | 95.4 |
| STS-E-1-2 | 1682 | 1938 | 95.4 | 1681 | 1939 | 95.4 |

| Interval | from | to | % |
|---|---|---|---|
| MESTS-b–a | 227 | 12 | 95.4 |

# 5 Discussion

## 5.1 Tsunami wave processes on the CTB and gravel deposits

Integrating the results of the incorporative hydrodynamic approach with the facies-stratigraphic analysis indicates that the depositions of both CTB and the earlier CT-2 events are attributed to tsunamis rather than typhoons.

Based on the size, shape, facies, elevation, and dating of the CTB, the wave process is thought to have lifted the CTB from the intertidal joint-bounded exposure, transported it via incipient saltation and terminal rolling over a total distance of over 30 m, and significantly overflowed the cliff (Fig. 2). As a result, a minimum wave height of 3.0 m (tsunami) or 12.0 m (typhoon) is estimated using the Froude Numbers 2.0 and 1.0, respectively. After calibrating the ancient sea level high and 100 year surge, the tallest typhoon waves (height = $0.73 \times$ water depth) that can form in the CT Channel are ruled out as insufficient for the CTB deposition. Typhoon waves reach only a maximum height of 3.3 m in the region, which would only cause the CTB to undergo slide–roll movement in the intertidal–subtidal zone (Table 3). The tallest typhoon waves in the supratidal zone of less than 1.5 m high are too weak to push or roll the CTB onto the 2.5 m high cliff. Moreover, the typhoon waves that are likely to form in the channel are probably overestimated because the modeled ratio of 0.73 for wave height to water depth at wave break appears to be excessive (Collins, 1970), as the ratio tends to be small, at around 0.44–0.6, in nearly horizontal bathymetry such as that observed in the Penghu area (Fig. 1B; Noormets et al., 2004). This is supported by the wave height maximum that was set by the 2015 Super typhoon Soudelor, which decreased from 6.8 m offshore in the 26-m-deep Paisha to 6.5 m in the 14-m-deep Makong Channel and 2.8 m in the 8-m-deep Longman at the nearshore (Fig. 1B; Liu et al., 2016). The ratio of 0.26–0.46 in these waterways is probably more applicable to the CT Channel. Accordingly, the tallest typhoon waves may have reached only 0.9–1.8 m in the CT Channel during the 16th–17th centuries, which is barely enough to initiate CTB sliding (Table 3).



The estimated wave period also associates CTB transport with long-periodicity tsunami or tsunami-like composite waves (Table 3). The tsunami wave origin appears to be dominant over the tsunami-like composite wave, according to the facies characteristics of poor sorting, rich suspension of fine-grained sediment, and matrix support (Figs. 3 and 6). The composite waves and induced currents usually have a low sediment capacity due to rapid attenuation following the wave break and low velocity as compared to tsunami waves (Watanabe et al., 2017; Dietrich et al., 2011; Bricker et al., 2014). Composite wave

breaks tend to result in multiple overwashes and sublayers that are characterized by normal sedimentary grading and moderate-to-high sorting (Baumann et al., 2017; Brill et al., 2016; Sohn and Sohn, 2019). These features of better sorting and stratification are absent in the ME gravel layers found in this study.

The supratidal elevation of the marine event gravels, 2.5–5.1 m a.s.l., also marks tsunami inundation with waves exceeding the 100 year typhoon surge of 1.8 m a.s.l. (Fig. 5a–b). Among the layers, MEWT and MESTS-b are lower at 2.1 and 2.0 m

a.s.l., respectively, and may have been deposited by 100 year surges in the 15th–17th centuries under the 0.2–0.3 m sea level high (Chen and Liu, 1996). However, the modern surge maximum that is associated with ongoing global warming may be too large to represent the surge conditions during the Little Ice Age in the 17th–19th centuries, because the historical surge of only 1.2–1.5 m a.s.l. is theoretically concordant with cool climate control (Hsu, 2007; Oey and Chou, 2016; Mei and Xie, 2016). Moreover, MEWT and MESTS-b are characterized by well-preserved articulated bivalves and gastropods, and feature

depositions from rapid catastrophic events rather than prolonged storm depositions with intense collision and abrasion between shells and rock clasts (Fig. 6B and D; Donato et al., 2008).

The estimated minimum tsunami waves that reach over 3.0 m in height significantly surpass the modern record of 0.4 m for the 1994 tsunami, which reached high onto the Penghu Islands. The wave height of several meters that is inferred from the paleotsunami deposits is not uncommon and has been recently reported both in the study area and in northern Taiwan (Yu et

al., 2016; Lu et al., 2019; Sugawara et al., 2019). Despite the absence of cliff-top boulders in the other outcrops, the supratidal elevation of marine gravel layers marks the minimum tsunami wave run-up (Pignatelli et al., 2009; Nanayama and Shigeno, 2006; Paris et al., 2018). It is noteworthy that the present layer elevation was likely 0.2–0.3 m lower during deposition in the 15th–17th centuries when the sea level was higher, and the beach profiles may shift landwards and upwards accordingly (Chen and Liu, 1996). Thus, the local wave height in the STS outcrop during the 17th-century CTB event is estimated at 2.4 m (Fig.

5a). The wave height during the 16th-century CT-2 event was probably 2.2 m at the CT-2 outcrop and 4.8 m at the SG outcrop (Fig. 5a–b).

**5.2 Probable event sources**

Because no historical earthquake and tsunami are recorded on Penghu Islands, it probably indicates far-field sources for the CTB and CT-2 events, such as the epicenter of the 1994 earthquake tsunami that was distant offshore to the SE (Fig. 1a).

Across the Taiwan Strait, three earthquakes with seawater overflows occurred in the period of the CT events, namely the 1604





and 1640 earthquakes in SE China and the 1661 earthquake in SW Taiwan (Keimatsu, 1963; Nakamura, 1935; Lei and Ou, 1991).

The newly identified 17th-century CTB tsunami is tentatively associated with the 1661 earthquake based on the historical accounts and the results of comparison with the previously reported forward simulations. According to the estimated wave
height of 2.4–4.0 m, the CTB tsunami was evidently much larger than the 0.4 m high tsunami that was associated with the 1994 M6.4 earthquake (Central Weather Bureau, 2022). The 1604 and 1661 earthquakes and overflows were probably more drastic and disastrous than the 1640 event (Keimatsu, 1963; Lei and Ou, 1991; Nakamura, 1935). The 1604 earthquake was previously associated with a shore-parallel NE–SW offshore rupture and an earthquake measuring M 8.0 near the SE China coast (Fig. 1A; Lei and Ou, 1991). The previously reported simulations of the earthquake mismatch the results of this study,
showing a marked cross-strait wave dissipation that varies from 0.96 m in height in SE China to 0.3 m in NW Taiwan (Huang et al., 2006; Wu, 2015). The 1661 earthquake was the largest on the Taiwan orogen that can be associated with compressional tectonics in the 17th century (Hsu, 2007; Nakamura, 1935). The previously reported simulations feature 2–4 m high tsunamis in SW Taiwan in association with scenarios of M8.2 earthquakes and reverse faulting and nonuniform slip distributions in the northern Manila Trench (Fig. 1A; Li et al., 2015). The estimated wave heights across SW Taiwan and the Penghu Islands are
comparable and thus ought to be included in future studies.

The 16th-century CT-2 tsunami was previously correlated to the 1604 earthquake (Lu et al., 2019), and is here tied to an unknown event based on the new dating result (1444–1575, 1510±65) that shows significant age discrepancy. The wave height derived from the layer at 2.2–4.8 m is comparable to the CTB tsunami wave height and likewise suggests a severe event that calls for more studies. Submarine landslide may be one of the top candidates, because potentially steep slopes are omnipresent
from the Penghu Islands to SW Taiwan and no major historical earthquakes are recorded across the Taiwan Strait in the period under question (Fig. 1a). Future simulation that includes the Penghu Islands is important, as the previously reported landslide scenarios show 5-m-high tsunamis in SW Taiwan (Li et al., 2015; Liu et al., 2022).

## 6 Conclusions

The coastal tsunami impact of the South China Sea was quantitatively evaluated for the first time by applying the incipient
motion formulas to the newly identified, 17th century, cliff-top boulder on Penghu Islands in the Taiwan Strait. The tallest typhoon waves that can form in the interisland bathymetry at the co-occurrence of ancient sea level high and 100 year surge are distinguished and found too weak to result in the 2.5 m high cliff-top deposition. The required tsunami height and period are estimated at 3.0 m and 5.6 h, respectively, rendering boulder deposition and terminal rolling possible. A local run-up wave height of 2.4–4.0 m is further inferred from the coeval gravel layers at three localities.
Compared to the other cliff-top boulders around the world (see Cox et al., 2020; Etienne et al., 2011; Goto et al., 2010 and references therein), the studied boulder is unique, and is characterized by the matrix-supported fabric, the upslope pinch-out bedform, and the lateral fining to pebble–cobble (Fig. 2). Together with the angular boulder shape, the boulder facies





records a suspension-rich turbulent flow process and the tsunami transport of bore-like waves from initial joint-bounded lift to saltation and rolling.

The new geological record obtained in this study improves our understanding of the South China Sea tsunamis and may contribute to future hydrodynamic simulation. The newly identified event in the 1575–1706 period is related to the disastrous 1661 earthquake in SW Taiwan and probably originated from megathrust activity in the northern Manila Trench (Fig. 1a). The previously reported event in the 15th–16th centuries is revised by the modeled age of 1444–1575 and is correlated to an unspecified event, of which submarine landslide may be a prominent candidate because of the widespread steep slopes in the

region (Lu et al., 2019).

**Data availability:**

The data are available upon request from the authors.

**Author contribution:**

NTY led the field investigation, calculated the hydrodynamic estimation, and drafted the manuscript. CHL and ICY

participated the field survey and CHL carried out the DEM image of CT Channel and boulder measurement. JHC undertook the grain size analysis. JYY and SJC provided important advises to the discussion and conclusions.

**Competing interests:**

The contact author has declared that none of the authors has any competing interests.

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
