# Peer review of "Boulder transport and wave height of a seventeenth century South China Sea tsunami on Penghu Islands, Taiwan"

_Natural Hazards and Earth System Sciences, 2022_

## Referee Comment (RC2)

Incipient motion formulas copied from Nandasena et al. (2022)

| Type of movement | Pre-setting conditions | | |
|---|---|---|---|
| | SB/SA | JBB | CEB |
| Sliding | $\dfrac{2C_{vv}(\rho_s/\rho_w - 1)gc(\mu_s\cos\theta + \sin\theta)}{C_{dv}(c/b) + \mu_s C_{lv}}$ | - | - |
| Overturning | $\dfrac{2C_{vv}(\rho_s/\rho_w - 1)gc(\cos\theta + (c/b)\sin\theta)}{C_{dv}(c^2/b^2) + C_{lv}}$ | - | $\dfrac{2C_{vv}(\rho_s/\rho_w - 1)gc}{C_{lv} - C_{dv}(c^2/b^2)}$ |
| Saltation/ lifting | $\dfrac{2C_{vv}(\rho_s/\rho_w - 1)gc(\cos\theta)}{C_{lv}}$ | $\dfrac{2C_{vv}(\rho_s/\rho_w - 1)gc(\cos\theta + \mu_s\sin\theta)}{C_{lv}}$ | $\dfrac{2C_{vv}(\rho_s/\rho_w - 1)gc}{C_{lv} - \mu_s C_{dv}(c/b)}$ |

Application of Fr number to calculate flow depth from flow velocity

$$F_r = \frac{u}{\sqrt{gh}}$$

$$h = \frac{u^2}{gF_r^2}$$

Therefore, the formulas for flow depth (wave height, h)

| Type of movement | Pre-setting conditions | | |
|---|---|---|---|
| | SB/SA | JBB | CEB |
| Sliding | $\dfrac{2C_{vv}(\rho_s/\rho_w - 1)c(\mu_s\cos\theta + \sin\theta)}{F_r^2[C_{dv}(c/b) + \mu_s C_{lv}]}$ | - | - |
| Overturning | $\dfrac{2C_{vv}(\rho_s/\rho_w - 1)c(\cos\theta + (c/b)\sin\theta)}{F_r^2[C_{dv}(c^2/b^2) + C_{lv}]}$ | - | $\dfrac{2C_{vv}(\rho_s/\rho_w - 1)c}{F_r^2[C_{lv} - C_{dv}(c^2/b^2)]}$ |
| Saltation/ lifting | $\dfrac{2C_{vv}(\rho_s/\rho_w - 1)c(\cos\theta)}{F_r^2 C_{lv}}$ | $\dfrac{2C_{vv}(\rho_s/\rho_w - 1)c(\cos\theta + \mu_s\sin\theta)}{F_r^2 C_{lv}}$ | $\dfrac{2C_{vv}(\rho_s/\rho_w - 1)c}{F_r^2[C_{lv} - \mu_s C_{dv}(c/b)]}$ |

For tsunami Fr = 2, for storms Fr = 1 (according to Nott's assumption)

The formulas given in the manuscript should be corrected if the authors referred to Nandasena et al. (2022).

Therefore, the formulas for storms should be given as:

**Your first equation has typo – you need to fix it as follow.**

| Type of movement | SB/SA | JBB |
|---|---|---|
| Sliding | $H \geq \dfrac{2C_{vv}(\rho_s/\rho_w - 1)c(\mu_s\cos\theta + \sin\theta)}{[C_{dv}(c/b) + \mu_s C_{lv}]}$ | - |
| Overturning | $H \geq \dfrac{2C_{vv}(\rho_s/\rho_w - 1)c(\cos\theta + (c/b)\sin\theta)}{[C_{dv}(c^2/b^2) + C_{lv}]}$ | - |
| Saltation/ lifting | $H \geq \dfrac{2C_{vv}(\rho_s/\rho_w - 1)c(\cos\theta)}{C_{lv}}$ | $H \geq \dfrac{2C_{vv}(\rho_s/\rho_w - 1)c(\cos\theta + \mu_s\sin\theta)}{C_{lv}}$ |

---

## Author Response (AR2)

**Point-by-point response:**

There are four comments from Referee 1 (RC1–4) and one comment from Referee 2 (RC5). The RC2 and 3 are the same, which makes a total of three comments from Referee 1 (RC1, 2, and 4).

**Response to RC1 of Referee 1:**

| Comment | Reply | Change in revised manuscript |
|---|---|---|
| (1) Lines 94-95: Why did you assume the wave height of the 100-year return period is 1.8 m? Have you done any probabilistic study for this assumption? Because Table 1 shows 100-year significant wave height is greater than 10 m. | Lines 94–95 (A modern surge maximum of 1.8 m a.s.l. is tentatively inferred to as the 100 year surge in this study.) are focused on the typhoon surge. The 50- and 100-year significant wave heights are presented in Table 1 and Lines 86–87.

There are previous probabilistic studies on the 50- and 100-year significant wave heights (see references in Table 1) and yet no previous probabilistic studies on the 100 year surge on the Penghu Islands. The 1.8 m a.s.l. is inferred from the modern 1.8 m surge maximum of the 2019 Typhoon Mitag among the 118 observed surges from 1997 to 2021 | The sentence is rephrased as 'In this study, the modern surge maximum of 1.8 m a.s.l. tentatively serves as an approximation for a 100-year surge.' (Lines 93–94). |

| | | |
|---|---|---|
| | (Lines 91–94). This surge maximum of the period of current global warming may be very close to the 100 year surge and comparable to the maximum in the 17th century of the Little Ice Age period. | |
| (2) Table 1 – what is "observation"? Is it the number of waves? | The 'observation' will be revised as 'number of measurements' to better label this column of the table that lists the total measurements at the selected buoys in certain months over the past 10 or 15 years. | Revised as 'number of measurements' in Table 1 (p. 5). |
| (3) Nandasena et al. (2022) formulas do not calculate the minimum wave height but the minimum flow velocity to initiate boulder transport. Therefore, the first four formulas given in the manuscript cannot be referred from Nandasena et al. (2022). The given formulas have a significant difference (perhaps typos) compared to the formulas in Nandasena et al. (2022). Therefore, the authors must include a section to explain how they derived their equations based on Nandasena et al. (2022). | The first four formulas in this study are the modified Nott's formulas of wave heights that were deduced from the flow velocity formulas. The flow velocity formulas were modified by a series of previous studies that took virtual boulder dimensions, maximum lifting surface, lift force, fundamental physics, effect of the bed slope, and transport mode sediment sources, transport distances, and shore slope angle into account (see references in Lines 60–64). | Revised in Line 166 of Table 2 (p. 10). |

| | In the revised manuscript, Nott (2003) and Nandasena (2020) are to be added to the source references of the four formulas in the footnote of Table 2. Nandasena (2020) reviewed most of the modifications except virtual boulder dimensions that were latter examined by Nandasena et al. (2022). | |
|---|---|---|
| (4) Line 135: Hudson formula is used for the design of armor-breakwaters against gravity waves (sea and swells). The formula was not validated for tsunamis and storms. However, Esteban et al. (2014) applied the Hudson formula to assess the damage to breakwaters by tsunamis. The authors may cite their paper to support the application of the Hudson formula in this study. | Hudson formula was only applied to the storm waves in this study (Tables 2 and 3). The application follows the study of Lorang (2011), which also used the formula and the modified Nott's formulas on storm wave estimates (Lines 135–136). | The sentence is rephrased as 'The Hudson formula is then adopted for independent storm wave estimates of the beach–intertidal zone (Lorang, 2011; Hudson, 1953).' (lines 139–140). |
| (5) Lines 136-137: the assumption of Fr =1 and 2 for storms and tsunamis, respectively, is outdated (comment 5.1). Because both the high-energy events can have similar Froude numbers varying from as small as 0.5 to as high as 2.5 or more. It is difficult to predict | From the authors' perspective, the use of fixed Froude numbers is not outdated and low in scientific value, and the suggestion of flow velocity may not be the best policy. In addition to Froude number, there are many other coefficients with limits in use in the | 1. Lines 131–134 are added for comment 5.3; The wave height/flow depth estimate is delineated a step further from flow velocity, as it has been deemed the most useful parameter in the analysis of ancient wave events and deposits (cf. Nandasena |

the exact Froude number at the pre-transport location of the boulders without knowing flow characteristics (flow depth and flow velocity). Therefore, the results based on this assumption have a low scientific value (comment 5.2). Alternatively, I suggest the authors to conclude based on flow velocities if permitted (5.3).

formulas that may results in uncertainties, which is well known to the authors and has been dealt with by numerous previously reported studies (Lines 58–65 and Sect. 3 Materials and methods). These responses have been agreed upon by the referee in his RC4, 'I am satisfied with the authors' responses... This is a good piece of work despite the limitations of the hydrodynamic formulas used in geo-science.'

Please see our previous responses in the interactive discussion online.

The responses are accommodated in the resubmitted manuscript (see the column on the right).

et al., 2022). Moreover, only the wave height records of historical and modern tsunamis and typhoons are available for comparison in the study area.

2. Lines 140–143 are added for comment 5.2; The Froude Number is set at 2.0 for tsunami waves and 1.0 for storm waves. The choice is based on the tendency of these waves to induce highly supercritical and critical flows, respectively (Nott, 2003). It is worth noting that various supercritical flow regimes are associated with both tsunami and storm waves (Cox et al., 2020; Nandasena, 2020), which will be addressed in Sect. 4.1.

3. New subsection 4.1.1 Storm wave height (line 212) and lines 220–226 are added for comment 5.2 to address the supercritical onshore flows induced by storm waves with the unfixed Froude Number between 1.0 and 1.6.

4. New subsection 4.1.2 Tsunami wave height (lines 227–262) are added for

| | | comment 5.2 to address the supercritical to critical flows induced by tsunami waves with the Froude Number between 1.0 and 2.0. |
|---|---|---|
| (6) Table 3: Some tsunami periods are highly unrealistic. For example, 3.4 S, and 3.6 S. Tsunamis are considered long-period waves. The calculated numbers fall in short-period waves. The authors need to declare which formulas used to calculate wave period (Lorang or Barbano) (6.1) and describe their results carefully following the established scientific definitions (6.2). | (Reply to 6.1) The formulas used to calculate wave periods **were already declared in Table 2**.

 (Reply to 6.2) The authors do agree that the tsunami waves with 3.4 and 3.6 s periods in the supratidal zone are undistinguishable from the storm waves. **It may indicate that the present formulas need to be improved to better estimate the tsunami waves in the supratidal setting, which is out of the scope of the present study.** Or the successive shortening of the estimated period in the intertidal–supratidal zone probably responds to the deceleration of the tsunami wave during shoaling that also causes a landward decrease in wavelength alongside an increase | The sentence is rephrased for comment 6.2 (lines 274–276); The successive shortening of the estimated period in the intertidal–supratidal zone likely corresponds to the deceleration of the tsunami waves through shoaling, which also causes a landward decrease in wavelength and a landward increase in wave height. |

| | in the wave height (**Table 2 and Lines 244–245**). | |
|---|---|---|

**References**

Central Weather Bureau: Wave Statistics, https://www.cwb.gov.tw/V8/E/C/MMC_STAT/sta_wave.html,   2022.

Chen, Y.-G. and Liu, T.-K.: Sea level changes in the last several thousand years, Penghu Islands, Taiwan Strait, Quat. Res., 45, 254–262, https://doi.org/10.1006/qres.1996.0026, 1996.

Cox, R., Ardhuin, F., Dias, F., Autret, R., Beisiegel, N., Earlie, C. S., Herterich, J. G., Kennedy, A., Paris, R., Raby, A., Schmitt, P., and Weiss, R.: Systematic review shows that work done by storm waves can be misinterpreted as tsunami-related because commonly used hydrodynamic equations are flawed, Front. Mar. Sci., 7, https://doi.org/10.3389/fmars.2020.00004, 2020.

Hudson, R. Y.: Wave forces on breakwaters, Trans. Am. Soc. Civil Eng., 118, 653–674, https://doi.org/10.1061/TACEAT.0006816, 1953.

Lorang, M. S.: A wave-competence approach to distinguish between boulder and megaclast deposits due to storm waves versus tsunamis, Mar. Geol., 283, 90–97, http://dx.doi.org/10.1016/j.margeo.2010.10.005, 2011.

Nandasena, N. A. K.: Chapter 29: Perspective of incipient motion formulas: boulder transport by high-energy waves, in: Geological records of tsunamis and other extreme waves, in: Geological records of tsunamis and other extreme waves, edited by: Engel, M., Pilarczyk, J., May, S. M., Brill, D., and Garrett, E., Elsevier, 641–659, https://doi.org/10.1016/B978-0-12-815686-5.00029-8, 2020.

Nandasena, N. A. K., Scicchitano, G., Scardino, G., Milella, M., Piscitelli, A., and Mastronuzzi, G.: Boulder displacements along rocky coasts: A new deterministic and theoretical approach to improve incipient motion formulas, Geomorphology, 407, https://doi.org/10.1016/j.geomorph.2022.108217, 2022.

Nott, J.: Waves, coastal boulder deposits and the importance of the pre-transport setting, Earth Planet. Sci. Lett., 210, 269–276, https://doi.org/10.1016/S0012-821X(03)00104-3, 2003.

**Response to RC2 of Referee 1:**

| Comment | Reply | Change in revised manuscript |
|---|---|---|
| The first formula has typos. Please follow the attached document. | The formula is revised accordingly in Table 2.

The new results from the corrected formula are listed in Table 3.

The interpretation and discussion are revised accordingly. | Revised in Tables 2 (the first formula; p. 9) and 3 (the results of sliding; p. 14) and lines 203–205. |

**Response to RC4 of Referee 1:**

| Comment | Reply | Change in revised manuscript |
|---|---|---|
| I am satisfied with the authors' responses and hope these revisions will be appeared in the final manuscript.

This is a good piece of work despite the limitations of the hydrodynamic formulas used in geo-science. | The authors are grateful for the kind and positive reply of the referee. The revisions in our previous replies will be included in the final version of the manuscript. | See the changes listed in the above two tables for RC1 and RC2. |

**Response to RC5 of Referee 2:**

| Comment | Reply | Change in revised manuscript |
|---|---|---|
| (1) The paper reports on matrix-supported boulder… On this basis alone I am convinced that these are tsunami deposits rather than typhoon deposits. | The authors are encouraged by the positive comment on one of the major contributions of the present study, i.e., presenting facies constraints on the sediment transport of the paleotsunami gravels and basalt boulders on the Penghu Islands. The key points summarized by the referee are comparable to **lines 26–27, 110–115, and 291–350/Sect. 4.2** of the submitted manuscript. | 1. The sentences in lines 26-27, 110-115, and 291–350 are rephrased by the Elsevier Language Editing Plus service. |
| (2) I also believe the authors should established a more solid comparison with similar deposits and their characteristics, as this is would strengthen the argumentation of the paper.

Perez-Torrado et al. 2006. The Agaete tsunami deposits (Gran Canaria): evidence of tsunamis related to flank collapses in the | a. One of the suggested references was already cited in the previously submitted manuscript, namely Paris et al. (2018) which described the tsunami deposits in Hawaii (Lines 300–302).
b. As suggested by the referee, Madeira et al. (2020) and Pérez-Torrado et al. (2006) are included, showing similar facies characteristics of the tsunami deposit studied. | See additions of the suggested references in lines 338 and 417. |

| | |
|---|---|
| Canary Islands. Mar. Geol. 227 (1–2), 137–149.

Paris et al., 2011. Tsunami deposits in Santiago Island (Cape Verde archipelago) as possible evidence of a massive flank failure of Fogo volcano. Sediment. Geol. 239, 129–145.

**Paris et al., 2018. Mega-tsunami conglomerates and flank collapses of ocean island volcanoes. Marine Geology, 395, pp.168-187.**

Ramalho et al., 2015. Hazard potential of volcanic flank collapses raised by new megatsunami evidence. Sci. Adv. 1 (2015), e1500456.

Madeira et al., 2020. A geological record of multiple Pleistocene tsunami inundations in an oceanic island: the case of Maio, Cape Verde. Sedimentology, 67(3), pp.1529-1552. | c. In the submitted manuscript, the tsunami deposit studied were compared with those reported on the Japan Sea and Pacific coasts of Hokkaido (Fujiwara and Kamataki, 2008; Nanayama and Shigeno, 2006). The common occurrences of articulated bivalves and stranded pumices in the tsunami deposits reported on the Pakistan coast (Lines 374–376; Donato et al., 2008) and on the northern coast of Taiwan (Lines 310–311; Yu et al., 2022) were also used for comparison. | |

| | | |
|---|---|---|
| (3) Below are a few passages of the text that I suggest revising, given that they are (in my view) confusing and not very clear, as well as a few minor language edits I suggest. | The authors are indebted to the referee for the editing advices. Most of them are accommodated and the responses and changes are here listed. | In addition to the referee's suggestions, the resubmitted manuscript has been edited by the Elsevier Language Editing Plus service. |
| Line 78 – what do you mean by "more than 90 units of Miocene basalt platform"? I really do not understand what the authors mean here... the term "units" in geology generally refers to stratigraphic units, yet I presume the authors here use the term with the meaning of individual boulders or clasts... so I suggest revising this to a more objective term – perhaps "more than 90 boulders derived from the Miocene basaltic platform"? | The units are replaced by volcanic islands. | The sentence is rephrased as 'The southern part of the strait comprises the Penghu Islands, which consist of more than 90 volcanic islands made of Miocene basalt.' (lines 75–77). |
| Line 81 – the authors describe sea-level fall... I presume they refer to local relative sea-level fall... is this correct? Please be more precise/objective and state if you refer to relative or eustatic sea level, and please provide more information as to the nature of | It is revised as 'local sea level'.  It was previously reported that the Holocene local sea level changes were dominated by the global sea level (eustatic) fluctuations due to the local tectonic quiescence. Please refer to Lines 80–84 and references therein. | The sentence is rephrased as 'Accordingly, the local sea level in this area has been controlled by the global sea level fluctuations, falling at approximately 5.1 cm per century from a 2.4 m highstand since 4.7 ka (Chen and Liu, 1996).' (lines 80–81). |

| | | |
|---|---|---|
| this sea-level change (climate-related? Subsidence/uplift related?) | | |
| Line 110 - another reference to largest boulder unit... again I presume the authors refer to a particular clast or boulder and not a unit composed of boulders... if so please remove the term "unit" from this phrase. | The units is replaced by 'clast'. | The sentence is rephrased as 'The largest boulder clast, hereafter referred to as 'Chungtun-1 boulder,' was selected for analysis due to its significantly larger size compared to the others in the outcrop (Fig. 3a).' (lines 110–111). |
| Line 111 – I find the following phrases really confusing "The cliff-top boulders are supported by a gravel and mud matrix that forms a lateral gravel layer (MECT-1) that pinches out from 2.5 to 4.0 m a.s.l. Marine shells and rounded pumice pebbles that are abundant in both matrix and gravel layer, and are also found on modern beaches in the region (Fig. 3b), are absent in the underlying basalt basement, basal soil, and overlying angular-gravel colluvium." Could you please reformulate these phrases and make it more concise? | The referee's suggestion is appreciated and will be adopted with a slight modification to feature the 'pinch-out' bedform that marks the minimum run-up level. | Lines 111–115 are rephrased as 'These boulder clasts form a cluster in a mud-matrix-supported gravel layer that is laterally continuous and gradually thins out upward from 2.5 to 4.0 m a.s.l. This layer is referred to as 'Chungtun-1 layer' (Fig. 3a). Marine shells and rounded pumice pebbles are abundant in the matrix and also on the channel beach (Fig. 3b); however, they are absent in the underlying basalt basement and basal soil, as well as the overlying angular-gravel colluvium. Accordingly, it is assumed that the cliff-top boulder and gravel layer have a marine sediment origin.' |

| | | |
|---|---|---|
| Here is a possible suggestion: "The cliff-top boulders are supported by a gravel and mud matrix that forms a laterally-continuous layer (MECT-1) with variable thickness and extending from 2.5 to 4.0 m a.s.l. Marine shells and rounded pumice pebbles are also abundant in the matrix, can be also found on modern beaches in the region (Fig. 3b), but are distinct from the underlying basalt basement, and are absent in the basal soil and overlying angular-gravel colluvium." | | |
| Lines 119-121 – change the existing phrase to "An intertidal rock exposure that is located 0.5 m below sea level and is covered by isolated and stacked boulders of sizes and shapes that are comparable to the CTB may be the source of the studied boulder (Fig. 3d)" | The sentence is rephrased. | Lines 120–122: The rephrased sentence; 'There is an intertidal rock exposure at 0.5 m below sea level, which exhibits well-developed joint fractures and rock debris similar in size and shape to the boulder clasts of the Chungtun-1 outcrop (Fig. 3a and d). |
| Lines 173-175 – this phrase is also very confusing... I suggest changing "better obtain" to "understand"... also, what do you mean by "during a marine event"? Are you | The word 'obtain' is replaced as suggested. The 'gravel layers that were deposited during a marine event' is revised as 'gravel layers that are associated with events of marine inundation and deposition'. | The sentence is rephrased as 'To obtain a better understanding of the facies and stratigraphic constraints on the transport and deposition of the cliff-top boulder, the Chungtun-1 outcrop and three additional |

| | | outcrops with gravel layers of marine sediment origin were investigated;' (lines 179–180). |
|---|---|---|
| referring to a storm? A typhoon? Another tsunami? please be more concise... | | |
| Lines 200-203 – similar to my comment to line 81, I find the statement "During this period, the maximum water depth in the CT Channel could increase from 2.5 to 4.5 m because the sea level was approximately 0.2 m higher than it is at present" confusing... first I would suggest changing "could increase from 2.5 to 4.5 m" to "was 2.5 to 4.5 m higher than today". Presumably you are also referring to relative sea-level change – could you be more precise/objective here? | The 'sea level' is revised as the 'local sea level'.

We tried to precisely express that the maximum water depth in the Chungtun Channel could increase from 2.5 to 4.5 m. It is 2.5 m, as measured in the fair-weather conditions by the authors and may be 4.5 m at a 100 year surge considering the higher local sea level in the 16th–17th centuries (Fig. 2a). | The sentence is rephrased as 'During this period, the maximum water depth in the Chungtun Channel could have reached 4.5 m because the local sea level was approximately 0.2 m higher than it is at present, allowing a 100-year surge of 1.8 m a.s.l. to occur (Fig. 2c; Chen and Liu, 1996; Central Weather Bureau, 2022).' (lines 211–213). |
| Lines 213-214 – I also find this phrase really confusing and grammatically incorrect: "The CTB is floored by the pumice-bearing gravel and mud matrix above the cliff basement (Fig. 4d) and the gravel layer are matrix-supported" – could you please revise this phrase to improve clarity? | | The sentence is rephrased as 'The boulder is underlain by the pumice-bearing gravel and mud matrix above the cliff basement (Fig. 4d) and evidently part of the matrix-supported Chungtun-1 gravel layer (Fig. 3a–b).' (lines 239–240). |

| Finally, in my view the text overuses acronyms/abbreviations, making it really difficult to read... | We appreciate the advice.

    The CTB for CT boulder is removed.

    The terms 'marine event' and 'ME' is removed.

    The coding and naming of the key gravel layers are removed.

    Only the common abbreviations are preserved, such as a.s.l. for above sea level. | Please see the changes in the resubmitted manuscript. |